# CHAIN-OF-EXPERTS: WHEN LLMS MEET COMPLEX OPERATIONS RESEARCH PROBLEMS

**Ziyang Xiao**[1], **Dongxiang Zhang**[1]*, **Yangjun Wu**[1], **Lilin Xu**[1], **Yuan Wang**[2],
**Xiongwei Han**[2], **Xiaojin Fu**[2], **Tao Zhong**[2], **Jia Zeng**[2], **Mingli Song**[1], **Gang Chen**[1]

[1] Zhejiang University   [2] Huawei Noah's Ark Lab
[3] School of Business, Singapore University of Social Sciences
{xiaoziyang, zhangdongxiang, brooksong, cg}@zju.edu.cn
yangjun.wu@connect.polyu.hk, {hanxiongwei, Zeng.Jia}@huawei.com
xulilin201@126.com, Jessicawang36@gmail.com
fuxiaojin32@hotmail.com, zhongtao5@huawei.com

## ABSTRACT

Large language models (LLMs) have emerged as powerful techniques for various NLP tasks, such as mathematical reasoning and plan generation. In this paper, we study automatic modeling and programming for complex operations research (OR) problems, so as to alleviate the heavy dependence on domain experts and benefit a spectrum of industry sectors. We present the first LLM-based solution, namely Chain-of-Experts (CoE), a novel multi-agent cooperative framework to enhance reasoning capabilities. Specifically, each agent is assigned a specific role and endowed with domain knowledge related to OR. We also introduce a conductor to orchestrate these agents via forward thought construction and backward reflection mechanism. Furthermore, we build a benchmark dataset (ComplexOR) of complex OR problems to facilitate OR research and community development. Experimental results show that CoE significantly outperforms the state-of-the-art LLM-based approaches both on LPWP and ComplexOR.

## 1 INTRODUCTION

Operations research (OR) aims to mathematically model complex decision-making problems that arise from a wide spectrum of industry sectors. To automate the procedure and reduce the dependence on domain-specific modeling experts, NL4Opt (Natural Language for Optimization) (Ramamonjison et al., 2022a) has recently emerged as an attractive but challenging NLP task. Its objective is to translate the text description of an OR problem into math formulations for optimization solvers. To facilitate understanding the task, an example from the current NL4Opt benchmark dataset is depicted in Figure 1. The prevailing NL4Opt models adopt a two-stage framework. Initially, they perform NER to identify variables, parameters, and constraints from the input text, which are subsequently converted into a mathematical optimization model. Despite their efficacy in elementary OR problems, these approaches fail in tackling complex real-world challenges.

In this paper, we study the automatic modeling and programming of complex OR problems derived from real-world industrial demands. As shown in Figure 1, their text descriptions often contain implicit constraints, posing a substantial interpretation challenge for existing NL4Opt solvers. For instance, the phrase "zero lead times", highlighted in green, conveys the absence of any time lag between production orders. Additionally, it is imperative to possess domain-specific knowledge to understand terminologies such as "backlogging", "carryover", and "lot-sizing". Finally, in contrast to the explicit input numbers in the simple example, complex OR problems exhibit an abundance of implicit variables that require specification from domain modeling experts. The magnitude of variables and constraints in these complex problems introduces formidable hurdles and results in a longer reasoning chain.

---

*Corresponding author.

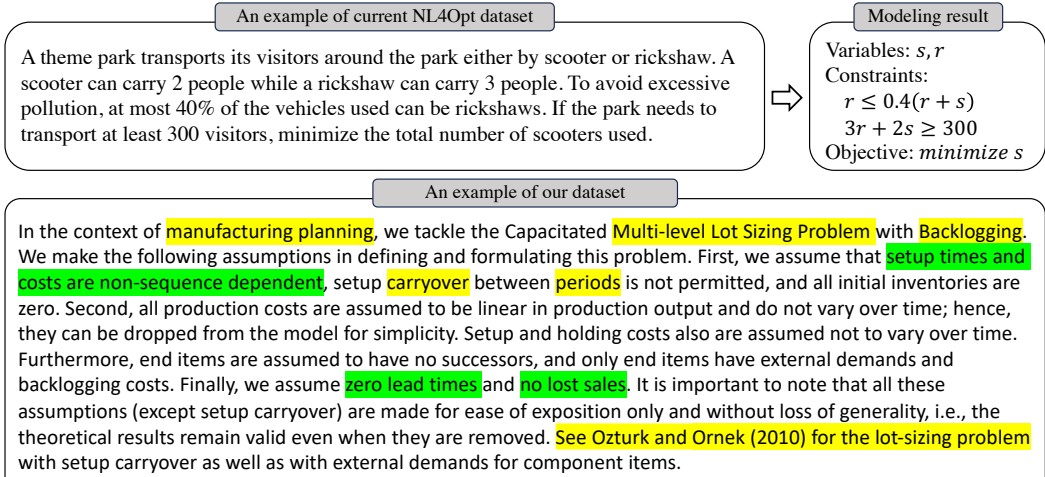

Figure 1: Comparison between elementary and complex NL4Opt problems. In the complex OR example, phrases in green indicate implicit constraints, and the Domain-specific terminologies are highlighted in yellow. The model output is presented in the Appendix A.1.

To resolve the above issues, we leverage the power of LLMs and present the first LLM-based solution. We propose a multi-agent reasoning framework, namely Chain-of-Experts (CoE), to orchestrate multiple LLM agents for complex OR problem solving. At the helm of the collaborative endeavor, there presides a central entity, designated as the "Conductor", responsible for orchestrating the sequence of interactions among the agents. Each agent is assigned a specific role and is equipped with domain-specific expertise. We implement diversified agents with different skills, including but not limited to terminology interpreter, construction of mathematical models and programming. Furthermore, we incorporate a backward reflection mechanism. Through a systematic analysis of the output, the framework has the capacity to detect potential errors in the problem-solving process.

**Comparison with other LLM-based reasoning.** In recent years, extensive research efforts have been devoted to enhancing the reasoning capabilities of Large Language Models (LLMs). Notable examples in this domain include Chain-of-Thought (Wei et al., 2022), Self-consistency (Wang et al., 2023a), Tree of Thoughts (Yao et al., 2023a), Graph of Thoughts (Besta et al., 2023), Progressive-Hint Prompting (Zheng et al., 2023), ReAct (Yao et al., 2023b). These works have formulated distinct prompting schemes and approaches to thought transformation. Further elaboration on these methodologies is presented in the subsequent section. Unfortunately, these single-agent LLMs as well as multi-agent schemes like Solo-Performance Prompting (Wang et al., 2023b) exhibit conspicuous limitations when confronted with complex OR problems because they cannot simultaneously tackle the challenges of implicit constraints, external knowledge prerequisites and long reasoning chain. In our CoE, we address these challenges via multi-expert collaboration and experimental results indicate that CoE can significantly outperform the LLM-based approaches.

**Contributions**. (1) We study NL4Opt at the more challenging level, which requires the model to have implicit constraint discovery, domain-specific knowledge, and complex reasoning capability. (2) This is the first LLM-based solution to complex OR problems. (3) We propose a novel multi-agent framework called Chain-of-Experts (CoE), enabling collaborative problem-solving and iterative modeling optimization based on the forward thought construction and backward reflection mechanism. (4) We also build a new dataset (ComplexOR) and the experimental results on it affirm the superior performance of CoE over 8 other LLM-based reasoning baselines.

## 2 RELATED WORK

**NL4Opt Problems.** NL4Opt aims to translate the descriptions of OR problems into mathematical formulations. A benchmark dataset[1] was curated by Ramamonjison et al. (2022a). To bridge the gap

---

[1]https://github.com/nl4opt/nl4opt-competition

between the natural language input $p$ and context-free formulation $r$, they proposed a two-stage mapping $p \rightarrow r \rightarrow f$ that first adopted the BART-base model (Lewis et al., 2020) with copy mechanism to generate an intermediate representation $r$, which was then parsed into a canonical formulation. Edit-based models (Malmi et al., 2022) can be applied as a post-processing step for error correction. The two-stage framework was followed by subsequent studies. He et al. (2022) introduced an ensemble text generator leveraging multitask learning techniques to enhance the quality of generated formulations. In a similar vein, Ning et al. (2023) proposed a prompt-guided generation framework, complemented by rule-based pre-processing and post-processing techniques, to enhance accuracy. In a related research endeavor, Prasath & Karande (2023) investigated the synthesis of mathematical programs. GPT-3 with back translation was utilized to synthesize the canonical forms as well as Python code.

**LLMs-based Reasoning.** Language models have shown substantial potential in solving complex reasoning tasks within specific domains, such as TSP(Zhang et al., 2023), databases(Xuanhe Zhou, 2023) and knowledge systems(Zhu et al., 2023). The Chain-of-Thought (CoT) (Wei et al., 2022) broke a complex reasoning task into a series of intermediate reasoning steps. Self-consistency (Wang et al., 2023a) replaced the greedy decoding in CoT by sampling a diverse set of reasoning paths and selecting the most consistent answer. Tree of Thoughts (ToT) (Yao et al., 2023a) and Graph of Thoughts (GoT) (Besta et al., 2023) further enhanced the reasoning capability by allowing LLMs to explore and combine thoughts in a structured manner. Progressive-Hint Prompting (PHP) (Zheng et al., 2023) progressively refined the answers by leverageing previously generated answers as hints. Subsequent works, such as ReAct (Yao et al., 2023b) and Reflexion (Shinn et al., 2023), allowed LLMs to interface with additional information or feedback from external sources. Recently, cooperation among multiple agents has also been explored. CAMEL (Li et al., 2023) introduced a novel communicative agent framework for autonomous cooperation. Solo Performance Prompting (SPP) (Wang et al., 2023b) transformed a single LLM into a cognitive synergist by simulating multiple personas and demonstrated the potential problem-solving abilities for multi-agent systems.

# 3 PROPOSED METHOD

## 3.1 EXPERT DESIGN

In our reasoning framework, an "expert" refers to a specialized agent based on a Large Language Model (LLM) augmented with domain-specific knowledge and reasoning skills. Each expert is assigned a specific role and undergoes four steps:

Step 1: **In-context Learning**. Each agent is allowed to access an external knowledge base and perform top-$k$ retrieval against the knowledge base. The retrieved information is then provided to the LLM to facilitate in-context learning. For example, an expert responsible for generating Gurobi programs can access the Gurobi official API documentation. This step is optional, depending on the availability of the knowledge base.

Step 2: **Reasoning**. LLM-based expert utilizes existing prompting techniques, such as Chain-of-Thought or self-consistency, to perform reasoning task according to their specific role. Our reasoning procedure consists of forward thinking and reflection modes, whose details will be presented in the subsequent section.

Step 3: **Summarize**. Due to the token limit constraint in a single interaction with LLM, an expert can choose to summarize their reasoning output. Since this step may result in significant information loss, it is optional for certain experts (e.g., modeling experts).

Step 4: **Comment**. This step is inspired by Solo Performance Prompting (Wang et al., 2023b), in which the participants are allowed to give critical comments and detailed suggestions. The objective is to make the communication between agents more constructive.

## 3.2 THE WORKFLOW OF CHAIN-OF-EXPERTS

The framework of our proposed Chain-of-Experts (CoE) is depicted in Figure 2. We initialize a collection of 11 experts such as terminology interpreter, modeling experts, programming experts, and code reviewing expert. Their detailed design specifications are available in Appendix A.2.1.

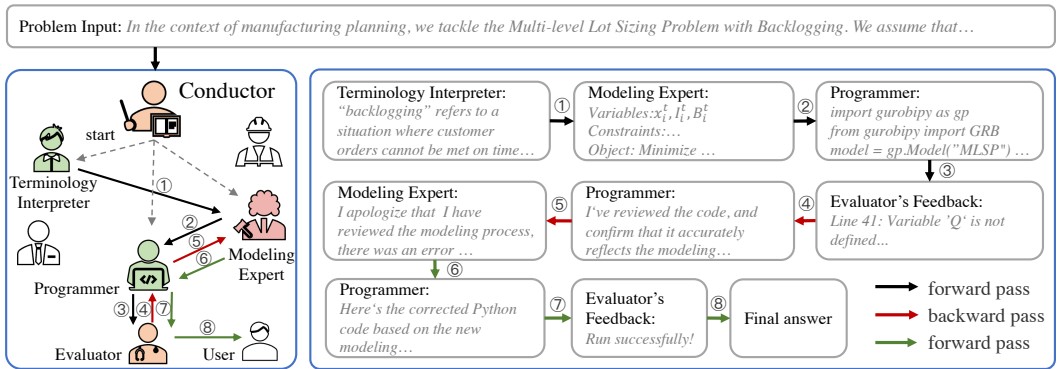

Figure 2: An example to illustrate the workflow of Chain-of-Experts. In this example, the Conductor receives the input problem and starts coordinating the experts. The exemplar workflow consists of ①: terminology interpretation for the input problem; ②: problem modeling; ③: program generation; ④: evaluation of correctness and identify an issue; ⑤: reflection of program, confirming correctness; ⑥: reflection modeling, find a mistake; ⑦: proceed with program generation again; ⑧: final evaluation, confirming correctness.

There is a **Conductor** to effectively coordinate these experts. It iteratively and dynamically selects an expert to construct a forward thought chain. The candidate answer generated by the forward reasoning chain will be passed to an external program execution environment, whose feedback signal triggers a backward reflection procedure. The details of these two key components are elaborated in the following:

**Forward Thought Construction.** During the forward-thought construction phase, the experts are sequentially selected by the Conductor. We can formulate forward thought construction as a sequential decision-making process, where we view the set of experts as the action space. Let's define the input problem description as $\mathcal{P}$, and a set of pre-defined experts as $\epsilon = \{E_{\phi_1}, E_{\phi_2}, ..., E_{\phi_n}\}$, where $n$ is the total number of experts and $\phi_i$ represents the configuration of the $i$-th expert. Each expert is associated with an optional knowledge base and a prompt template. We denote the set of comments at the $t$-th reasoning step as $\mathcal{C}_t$ and define the state in Equation 1.

$$S_t = (\mathcal{P}, \mathcal{C}_t, t) \tag{1}$$

Unlike traditional reinforcement learning which requires a large amount of training data, we utilize a training-free approach by leveraging the prompt technique of large language models. This allows us to achieve the same functionality as a decision-making agent without any training. Consequently, we model the policy of Conductor in Equations 2, where the Conductor acts as a policy function to select the experts, $\mathcal{F}$ represents the large language model, $\theta'$ represents the parameters of LLM, and $\mathbb{PT}_t$ denotes the prompt template at the $t$-th step.

$$Conductor_{\mathcal{F}^{\theta'}(\mathbb{PT}_t)}(e|s) = P_r\{E_{\phi_t} = e|S_t = s\} \tag{2}$$

Based on the above formulation, the expert selection policy can be translated into the design of a prompt template $\mathbb{PT}_t$, which requires prompt engineering to achieve an optimal policy. The detailed design of the prompt template is presented in Appendix A.2.2. After designing Conductor, we can update the comment set in each reasoning step as follows.

$$E_{\phi_{i_t}} = Conductor(S_t) \tag{3}$$

$$c = E_{\phi_{i_t}}(\mathcal{P}, \mathcal{C}_t) \tag{4}$$

$$\mathcal{C}_{t+1} = \mathcal{C}_t \cup \{c\} \tag{5}$$

where $E_{\phi_{i_t}}$ represents the selected $i_t$-th expert at step $t$ and $c$ denotes the comment of the selected expert. We concatenate the previous comments $\mathcal{C}_t$ and $c$ to obtain $\mathcal{C}_{t+1}$ as the new state. After a fixed number of steps $T$, the forward process in the Chain-of-Experts framework is terminated. At this point, all the comments are summarized to form the final answer $\mathcal{A}$.

**Backward Reflection.** The backward reflection mechanism in the Chain-of-Experts enables the system to leverage external feedback and adjust the collaboration among experts based on the

---

**Algorithm 1** Chain-of-Experts

---

**Input**: problem description $p$
**Parameters**: forward steps $N$, maximum forward-backward trials $T$

1: Initialize a set of comments $C \leftarrow \emptyset$
2: Initialize a stack of experts $E \leftarrow \emptyset$
3: **for** $t = 1, ..., T$ **do**
4:     **for** $i = 1, ..., N$ **do**
5:         $expert_i \leftarrow Conductor(p, C)$
6:         $comment \leftarrow expert_i(p, C)$
7:         $C \leftarrow C \cup \{comment\}$
8:         $E.push(expert_i)$
9:     **end for**
10:    $answer \leftarrow Reducer(p, C)$
11:    $feedback, passed \leftarrow Evaluator(answer)$
12:    **if** $passed$ **then**
13:        **return** $answer$
14:    **end if**
15:    $stop\_backward \leftarrow$ **false**
16:    **while not** $stop\_backward$ **and not** $E.empty()$ **do**
17:        $expert \leftarrow E.pop()$
18:        $feedback, stop\_backward \leftarrow expert.reflect(p, C, feedback)$
19:        $C \leftarrow C \cup \{feedback\}$
20:    **end while**
21: **end for**
22: **return** $answer$

---

evaluation of problem-solving results. Let's define the trajectory of experts selected in order as $\tau = \{E_{\phi_{i_1}}, E_{\phi_{i_2}}, ..., E_{\phi_{i_T}}\}$, where $i_t$ represents the index of the expert selected at step $t$. The backward reflection process starts with external feedback $r_{raw}$, which is typically provided by a program execution environment. This process can be denoted as $r_{raw} \leftarrow execution(\mathcal{A})$. Then, the initial signals are derived from the evaluation of the raw external feedback: $(r_0, sr_0) \leftarrow evaluate(r_{raw})$, where $r_0$ is a boolean value indicating whether the backward process needs to continue and $sr_0$ represents the natural language summary of the feedback, which is used to locate errors during the backward reflection process. If the answer $\mathcal{A}$ is deemed as correct, $r_0$ is set to false and the whole reasoning procedure terminates. Otherwise, the Chain-of-Experts initiates a backward self-reflection process to update the answer. The process begins with the last expert $E_{\phi_{i_T}}$, and backpropagates in reverse order to iteratively update the feedback signal. At the $t$-th backward step, the update of the state is described by Equation 6 and 7, where $reflect$ represents one of the reasoning abilities in expert design. The $reflect$ function also produces a tuple of $r_t$ and $sr_t$, which aligns with the output of the $evaluate$ function. In this case, $r_t$ is set to true when the expert confirms the correctness of its previous comment.

$$(r_t, sr_t) \leftarrow reflect(E_{\phi_{i_{T-t+1}}}, \mathcal{P}, \mathcal{C}_t, r_{t-1}) \tag{6}$$

$$\mathcal{C}_{t+1} = \mathcal{C}_t \cup \{sr_t\} \tag{7}$$

The backward reflection process continues until the feedback signal $r_t$ indicates that the expert $E_{\phi_{i_{T-t+1}}}$ is the one who made the mistake or or until all experts have been reflected upon. Subsequently, a forward process will be performed again.

It is worth noting that our reflection method differs from Reflexion (Shinn et al., 2023), where reflection is performed at the system level with interaction among multiple experts, and the feedback is recursively backpropagated. In contrast, Reflexion just involves a single LLM.

### 3.3 IMPLEMENTATION DETAILS

Algorithm 1 provides the implementation pseudo-code of the Chain-of-Expert framework, which consists of four main stages:

Initialization (lines 1 - 2): The process begins by initializing the set of comments $C$. Additionally, a stack $S$ is used to store the selected experts, ensuring a first-in-last-out order for forward thought construction and backward reflection.

Forward Thought Construction (lines 4 - 9): Experts are selected sequentially by the Conductor, with each expert contributing their comments to the global comment set $C$. Forward construction process continues for a fixed number of steps $N$. As depicted in line 10, once the forward construction is completed, a Reducer is employed to summarize all the comments and generate a final answer. Since the comment set contains numerous comments after the forward construction, the Reducer plays a crucial role in summarizing and reconciling these comments. For more detailed prompt template design of the Reducer, please refer to Appendix A.2.3.

Backward Reflection (lines 11 - 20): In line 11, once a solution is obtained, an Evaluator gathers feedback and converts it into natural language feedback to assess its correctness. If the solution is deemed incorrect, the system enters a reflection phase. In this phase, experts are consulted iteratively in reverse order by removing them from the stack. They are prompted to reflect on their solution and provide additional comments if necessary. As indicated in line 16, the backward process continues until a mistake is found by self-reflection or the first expert is reached.

Iterative Improvement (loop in line 3): The steps of forward thought construction and backward reflection are repeated iteratively until a satisfactory solution is achieved or a maximum number of trials $T$ is reached.

## 4 EXPERIMENTS

### 4.1 DATASETS

**LPWP**. The LPWP dataset (Ramamonjison et al., 2022b) is collected from the NL4Opt competition in NuerIPS 2022. It comprises 1101 elementary-level linear programming (LP) problems. Each problem consists of a text description with IR annotations including parameters, variables, linear constraints and the objective function. The dataset is partitioned into 713 training samples, 99 validation samples, and 289 test samples for performance evaluation.

**ComplexOR**. With the assistance from three specialists with expertise in operations research, we constructed and released the first dataset for complex OR problems. We selected 37 problems from diversifed sources, including academic papers, textbooks, and real-world industry scenarios. These problems cover a wide range of subjects, spanning from supply chain optimization and scheduling problems to warehousing logistics. It took the experts nearly a month to annotate each problem with model formulation, and a minimum of five test cases to verify the correctness of generated code.

### 4.2 MODEL SETUP AND PERFORMANCE METRICS

In our experimental setup, we use the GPT-3.5-turbo as the default large language model. We set the parameter *temperature* to a value of $0.7$ and conduct five runs to average the metrics. The number of iterations is set to 3, with each iteration consisting of 5 forward steps by default.

Since it is infeasible for domain experts to manually evaluate the output from the LLM-based solutions, we employed an automated code evaluation process. Specifically, we require each solution to generate the programming code for each OR problem. If the code can pass the associated test cases annotated by the OR specialists, we consider the problem is successfully solved. We use **Accuracy** to indicate the success rate. Besides this rigorous metric, we also adopt compile error rate **(CE rate)** to capture the percentage of generated programs that fail to compile, possibly caused by issues in the automatic modeling process; Alternatively, runtime error rate **(RE rate)** measures the percentage of generated programs that encounter errors during execution, which are caused by internal logic errors such as unsolvable models or non-linear constraints. The experimental code is at https://github.com/xzymustbexzy/Chain-of-Experts.

### 4.3 BASELINES

We compare CoE with 9 baselines. As to traditional approaches for NL4Opt, we consider **tag-BART** (Neeraj Gangwar, 2022) as a SOTA model, which won 1st place in the NeurIPS competition (Ramamonjison et al., 2022b). We also compare CoE with prevailing LLM-based methods, including **Chain-of-Thought**, **Progressive Hint**, **Tree-of-Thought**, **Graph-of-Thought**, **ReAct**, **Reflexion** and **Solo Performance Prompting**. The default GPT without any optimization on the reasoning

Table 1: Comparison with baselines on LPWP and ComplexOR

| Method | LPWP | | | ComplexOR | | |
|---|---|---|---|---|---|---|
| | Accuracy↑ | CE rate↓ | RE rate↓ | Accuracy↑ | CE rate↓ | RE rate↓ |
| tag-BART | 47.9% | - | - | 0% | - | - |
| Standard | 42.4% | 18.1% | 13.2% | 0.5% | 36.8% | 8.6% |
| Chain-of-Thought | 45.8% | 20.5% | 9.4% | 0.5% | 35.3% | 8.6% |
| Progressive Hint | 42.1% | 19.4% | 10.3% | 2.2% | 35.1% | 13.5% |
| Tree-of-Thought | 47.3% | 17.4% | 9.7% | 4.9% | 31.4% | 7.6% |
| Graph-of-Thought | 48.0% | 16.9% | 9.1% | 4.3% | 32.4% | 8.1% |
| ReAct | 48.5% | 15.5% | 11.2% | 14.6% | 31.9% | 10.8% |
| Reflexion | 50.7% | 7.3% | 9.0% | 13.5% | 12.9% | 10.1% |
| Solo Performance | 46.8% | 17.9% | 13.6% | 7.0% | 46.5% | 13.5% |
| CoE without expert | 55.1% | 4.0% | 11.9% | 18.8% | 7.9% | 15.0% |
| Chain-of-Experts | **58.9%** | **3.8%** | **7.7%** | **25.9%** | **7.6%** | **6.4%** |

chain is named **Standard**, which is expected to achieve inferior performance. We also implement a baseline that uses the same model, which uses a uniform system prompt, "You are a helpful assistant," across all roles, without any additional knowledge bases. The detailed implementation of these algorithms is described in Appendix A.3.

## 4.4 OVERALL PERFORMANCE ON LPWP AND COMPLEX OR

The results in terms of accuracy, CE rate and RE rate in the two benchmark datasets are reported in Table 1. Since the traditional method tag-BART is not capable of generating code, we measure its accuracy by requiring its constraints and objective in the math model to be correct. Note that generating a valid linear programming model is a prerequisite step for correct code generation. Even under such a loose evaluation metric, tag-BART is still inferior to certain LLM-based baselines, verifying the feasibility of applying LLM to solve OR problems. We also observe that tag-BART fails in all problem instances of ComplexOR.

Among the LLM-based baselines, Reflexion stands out as the most promising OR problem solver. Due to its self-reflection mechanism, it achieves the lowest CE rate and RE rate in both datasets. When confronted with complex OR problems, its overall accuracy is slightly inferior to ReAct. The reason is that in complex OR problems, the ability to access external knowledge bases becomes more crucial, which is a strength of ReAct. Even though Solo Performance also adopts a multi-agent reasoning framework, its performance is not satisfactory. Unlike our collaborative reasoning framework, its agents are simply initialized by a leader LLM and lack effective cooperation to solve the challenging OR problems.

Our proposed CoE established clear superiority among all performance metrics in both datasets. In LPWP, the accuracy is 58.9%, surpassing the state-of-the-art agent Reflexion by 8.2%. In its best performance, CoE also manages to solve 10 out of 37 complex problem instances in the ComplexOR. The outstanding performance owes to the effective design of the Chain-of-Experts reasoning framework, including the expert design methodology, the roles of the Conductor and Reducer, and the reflection mechanism. We also find that the removal of experts' features leads to a decrease in accuracy, which suggests that the CoE benefits from using specialized experts. In the next experiment, we will investigate the effect of these ingredients through an ablation study.

## 4.5 ABLATION STUDY

Regarding single expert design, Table 2 highlights the positive roles played by both the knowledge base and reasoning ability. Removing these components results in a slight drop in accuracy. Interestingly, we found that summarization is the most crucial design aspect. Reasons are two-fold. First, the length of comments may exceed the token limit of GPT-turbo-3.5 and the overflowed tokens will be discarded. Second, a compact and meaningful summary is more friendly for decision making in the downstream experts.

Table 2: Ablation study of Chain-of-Experts

| | Method | LPWP | | | ComplexOR | | |
|---|---|---|---|---|---|---|---|
| | | **Accuracy** | **CE rate** | **RE rate** | **Accuracy** | **CE rate** | **RE rate** |
| | CoE (Full) | 58.9% | 3.8% | 7.7% | 25.9% | 7.6% | 6.4% |
| inner-agent | w/o knowledge base | 58.0% | 4.0% | 8.5% | 23.3% | 8.4% | 7.9% |
| | w/o CoT reasoning | 58.2% | 3.7% | 7.9% | 24.3% | 8.1% | 6.4% |
| | w/o summarize | 56.3% | 3.8% | 9.4% | 20% | 7.6% | 10.3% |
| inter-agent | w/o Reflection | 55.6% | 4.2% | 12.2% | 22.7% | 7.8% | 10.6% |
| | w/o Conductor | 54.2% | 6.5% | 8.2% | 21.1% | 8.1% | 8.6% |
| | w/o Reducer | 56.5% | 5.5% | 8.8% | 23.0% | 9.2% | 8.1% |

For inter-expert collaboration, we evaluate the effect of backward reflection, forward thought construction in Conductor, and reducer. As shown in Table 2, after removing the component of backward reflection, the accuracy drops significantly from 58.9% to 55.6%, and the RE rate increases noticeably from 7.7% to 12.2%. These results imply that without the reflection mechanism, the system is prone to mistakes in logical reasoning and lacks the ability for self correction. To evaluate the effect of Conductor, we replace it with random selection of subsequent experts during the construction of the forward chain of thought. The performance also degrades significantly because the experts are no longer well coordinated and the random selection of experts may even be detrimental to the reasoning process. It's surprising to find that the Reducer component also contributes remarkably. This module summarizes the collective insights from multiple preceding experts. If we remove it, the answer will be extracted from the concatenation of the experts' raw comments, which may lead to incorrect conclusions, as the comments can be fragmented and even conflicting with each other.

## 4.6 PARAMETER SENSITIVITY ANALYSIS

We evaluate two critical parameters related to reasoning capabilities, including the number of steps in the forward thought construction and the *temperature* in the Large Language Model. From the results shown in Figure 3a, we can draw two conclusions. Firstly, a lower value for the *temperature* parameter tends to lead to better performance. This suggests that, for knowledge-intensive problems, the experts benefit from providing more deterministic and consistent thoughts, rather than creative or diverse ones. Secondly, a longer reasoning chain that involves more experts in the forward thought construction generally improves accuracy. However, it occurs at the cost of higher reasoning time cost and more API requests. That's why we select $temperature = 0$ and $forward\ steps = 5$ as the default parameter configuration.

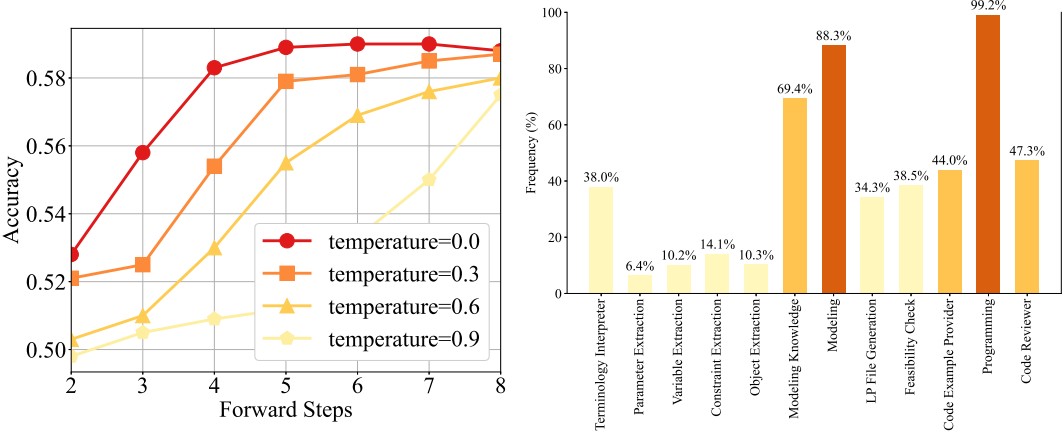

(a) CoE performance on different parameter settings.  (b) Selection frequency of individual expert.

Figure 3: Parameter sensitive analysis and selection frequency analysis on LPWP

Table 3: Robustness of Chain-of-Experts under different large language models

| Method | GPT-3.5-turbo | | GPT-4 | | Claude2 | |
|---|---|---|---|---|---|---|
| | **LPWP** | **ComplexOR** | **LPWP** | **ComplexOR** | **LPWP** | **ComplexOR** |
| Standard | 42.4% | 0.5% | 47.3% | 4.9% | 44.9% | 0.0% |
| Reflexion | 50.7% | 13.5% | 53.0% | 16.8% | 51.4% | 12.4% |
| Chain-of-Experts | **58.9%** | **25.9%** | **64.2%** | **31.4%** | **62.0%** | **27.0%** |

## 4.7 OTHER LLMs AS BASE REASONING MODEL

We also conduct an investigation into the impact of using different LLMs within the Chain-of-Experts. We consider **GPT-4** and **Claude2** as two alternative LLMs and select **Standard** and **Reflexion** as two baselines. As shown in Table 3, all methods benefit from the upgrade of more advanced LLMs. However, our Chain-of-Experts approach exhibited the most substantial improvement. For instance, when GPT-4 is used, we observed an accuracy boost of 8.3% in LPWP and 5.5% in ComplexOR, the highest among the three methods.

## 4.8 EXPERIMENTAL ANALYSIS OF EXPERT SELECTION FREQUENCY

In the final experiment, we aim to gain a deeper understanding of the Conductor's behavior and examine the rationality behind its selection of experts.

First, we conduct experiments on the LPWP dataset and analyze the selection frequency of each expert. In Figure 3b, we observe that the Programming Expert and Modeling Expert are the two most frequently selected experts. This finding is consistent with the expectation that modeling and programming are crucial to solving OR problems. Additionally, we notice that the extraction of parameters, variables, constraints, and objective functions is rarely selected. This can be attributed to advancements in language comprehension capabilities of LLMs, which now can understand problem statements directly, without the need for the step-by-step NER in traditional methods.

Moreover, we study the most frequently sampled collaboration paths involving multiple experts. Each problem-solving process provides a path that represents the order of experts involved. In Table 4, we observe that when the parameter *forward steps* is set to 2, where only two experts collaborate to solve a problem, the most frequent path is from the Modeling to the Programming Expert. This finding aligns with the importance of these two roles in problem-solving. Additionally, when the steps is set to 6, the collaboration path becomes more complex and resembles real-world workflows.

Table 4: The most frequently collaboration paths for experts on different forward steps settings.

| Forward steps | Most frequent path |
|---|---|
| 2 | Modeling → Programming |
| 3 | Knowledge → Modeling → Programming |
| 4 | Terminology Interpreter → Knowledge → Modeling → Programming |
| 5 | Terminology Interpreter → Modeling → LP file Generator → Programming → Code Reviewer |
| 6 | TI → Modeling → LP file Generator → Programming Example Provider → Programming → Code Reviewer |

## 5 CONCLUSION

In this paper, we presented the first LLM-based solution to complex OR problems. To enhance reasoning capabilities, we devised Chain-of-Experts (CoE), a novel multi-agent cooperative framework. The core of CoE was a conductor orchestrating a group of LLM-based experts via forward thought construction and backward reflection mechanism. We built a new dataset, ComplexOR to facilitate OR research and community development. Experimental results indicated that our CoE significantly outperformed the state-of-the-art reasoning methods both on LPWP and ComplexOR.

## 6 ACKNOWLEDGEMENTS

The work is supported by the National Key Research and Development Project of China (2022YFF0902000).

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

## A APPENDIX

### A.1 AN EXAMPLE OF COMPLEXOR DATASET

The example shown in Figure 4 shows a typical instance of a complex operations research problem. Specifically, it illustrates a Capacitated Lot Sizing Problem, which is much more challenging. This problem involves a wide range of constraints, such as summations, equations, and inequalities. Unlike simpler problems, the objective function in this case is not a straightforward linear expression but rather a summation across multiple set variables. These combined characteristics categorize this problem as a complex optimization challenge.

An example of our dataset

In the context of manufacturing planning, we tackle the Capacitated Multi-level Lot Sizing Problem with Backlogging. We make the following assumptions in defining and formulating this problem. First, we assume that setup times and costs are non-sequence dependent, setup carryover between periods is not permitted, and all initial inventories are zero. Second, all production costs are assumed to be linear in production output and do not vary over time; hence, they can be dropped from the model for simplicity. Setup and holding costs also are assumed not to vary over time. Furthermore, end items are assumed to have no successors, and only end items have external demands and backlogging costs. Finally, we assume zero lead times and no lost sales. It is important to note that all these assumptions (except setup carryover) are made for ease of exposition only and without loss of generality, i.e., the theoretical results remain valid even when they are removed. See Ozturk and Ornek (2010) for the lot-sizing problem with setup carryover as well as with external demands for component items.

⇩

Modeling Result

Sets: $Perios, M, I, end, eta$

Parameters: $sc, bc, hc, st, a, gd, Mn, r, C$

Variables: $x, s, b, y, bc$

Constraints:

invBalance1: $x_{i,t} + s_{i,t-1} + b_{i,t} - b_{i,t-1} = gd_{i,t} + s_{i,t}$ for $i \in end, t \in Periods$

invBalance2: $x_{i,t} + s_{i,t-1} = gd_{i,t} + \sum_{j \in eta_i} r_{i,j} \cdot x_{j,t} + s_{i,t}$ for $i \in I - end, t \in Periods$

capConstraints: $\sum_{i \in I} a_{i,m} \cdot x_{i,t} + \sum_{i \in I} st_{i,m} \cdot y_{i,t} <= C_{m,t}$ for $m \in M, t \in Periods$

setupConstraints: $x_{i,t} <= Mn \cdot y_{i,t}$ for $i \in I, t \in Periods$

initStatus: $s_{i,0} = 0$ for $i \in I$

endStatus: $b_{i,T} = 0$ for $i \in I$

Objective: $minimize \sum_{i \in I, t \in Periods}(sc_i \cdot y_{i,t} + hc_i \cdot s_{i,t}) + \sum_{i \in end, t \in Periods} bc_i \cdot b_{i,t}$

Figure 4: An example of ComplexOR dataset.

## A.2 MORE IMPLEMENTATION DETAILS OF CHAIN-OF-EXPERTS

### A.2.1 EXPERTS DESIGN

In this section, we provide an in-depth overview of the individual experts participating in the Chain-of-Experts framework. Table 5 offers a comprehensive list of these experts, each assigned a specific role and domain knowledge relevant to OR problem-solving.

Table 5: All experts involved in Chain-of-Experts

| Expert name | Knowledge base |
|---|---|
| Terminology Interpreter | Supply Chain Optimization & Scenario Modeling |
| Parameter Extraction Expert | - |
| Variable Extraction Expert | - |
| Constraint Extraction Expert | - |
| Objective Function Extraction Expert | - |
| Modeling Knowledge Supplement Expert | GAMS-Cutting Edge Modeling |
| Modeling Expert | - |
| LP File Generation Expert | LP format Documentation |
| Constraint Feasibility Check Expert | - |
| Programming Example Provider | Gurobi Example Tour |
| Programming Expert | Gurobi Reference Manual |
| Code Reviewer | - |

Below, we present the detailed descriptions and prompt template implementations for each expert. Please note that text enclosed within curly braces signifies placeholders that will be dynamically populated during runtime based on the specific problem description, comments provided by experts, and retrieved knowledge.

**Terminology Interpreter:**

**Role description**: Provides additional domain-specific knowledge to enhance problem understanding and formulation.

**Prompt template**: As a domain knowledge terminology interpreter, your role is to provide additional information and insights related to the problem domain. Here are some relevant background knowledge about this problem: $\{knowledge\}$. You can contribute by sharing your expertise, explaining relevant concepts, and offering suggestions to improve the problem understanding and formulation. Please provide your input based on the given problem description: $\{problem\}$.

### Parameter Extraction Expert:

**Role description**: Provides additional domain-specific knowledge to enhance problem understanding and formulation.

**Prompt template**: As a variable extraction expert, your role is to identify and extract the relevant variables from the problem statement. Variables represent the unknowns or decision variables in the optimization problem. Your expertise in the problem domain will help in accurately identifying and describing these variables. Please review the problem description and provide the extracted variables along with their definitions: $\{problem\}$.

### Variable Extraction Expert:

**Role description**: Proficient in identifying and extracting relevant variables from the problem statement.

**Prompt template**: As a variable extraction expert, your role is to identify and extract the relevant variables from the problem statement. Variables represent the unknowns or decision variables in the optimization problem. Your expertise in the problem domain will help in accurately identifying and describing these variables. Please review the problem description and provide the extracted variables along with their definitions: $\{problem\}$.

### Constraint Extraction Expert:

**Role description**: Skilled in extracting constraints from the problem description.

**Prompt template**: As a constraint extraction expert, your role is to identify and extract the constraints from the problem description. Constraints represent the limitations or conditions that need to be satisfied in the optimization problem. Your expertise in the problem domain will help in accurately identifying and formulating these constraints. Please review the problem description and provide the extracted constraints: $\{problem\}$. The comments given by your colleagues are as follows: $\{comments\}$, please refer to them carefully.

### Objective Function Extraction Expert:

**Role description**: Capable of identifying and extracting the objective function from the problem statement.

**Prompt template**: You are an expert specialized in Operations Research and Optimization and responsible for objective function extraction. Your role is to identify and extract the objective function from the problem statement. The objective function represents the goal of the optimization problem. Now, the problem description is as following: $\{problem\}$.

### Modeling Knowledge Supplement Expert:

**Role description**: Offers supplementary knowledge related to modeling techniques and best practices.

**Prompt template**: As a modeling knowledge supplement expert, your role is to provide additional knowledge and insights related to modeling techniques and best practices in the field of Operations Research and Optimization. Here are some relevant background knowledge about modeling technique: $\{knowledge\}$. You can contribute by explaining different modeling approaches, suggesting improvements, or sharing relevant tips and tricks. Please provide your input based on the given problem description and the modeling efforts so far: $\{problem\}$.

**Modeling Expert:**

**Role description**: Proficient in constructing mathematical optimization models based on the extracted information.

**Prompt template**: You are a modeling expert specialized in the field of Operations Research and Optimization. Your expertise lies in Mixed-Integer Programming (MIP) models, and you possess an in-depth understanding of various modeling techniques within the realm of operations research. At present, you are given an Operations Research problem, alongside additional insights provided by other experts. The goal is to holistically incorporate these inputs and devise a comprehensive model that addresses the given production challenge. Now the origin problem is as follow: $\{problem\}$. And the modeling is as follow: $\{comments\}$ Give your model of this problem.

**LP File Generation Expert:**

**Role description**: Expertise in generating LP (Linear Programming) files that can be used by optimization solvers.

**Prompt template**: As an LP file generation expert, your role is to generate LP (Linear Programming) files based on the formulated optimization problem. LP files are commonly used by optimization solvers to find the optimal solution. Here is the important part source from LP file format document: $\{knowledge\}$. Your expertise in generating these files will help ensure compatibility and efficiency. Please review the problem description and the extracted information and provide the generated LP file: $\{problem\}$. The comments given by your colleagues are as follows: $\{comments\}$, please refer to them carefully.

**Programming Example Provider:**

**Role description**: Provides programming examples and templates to assist in implementing the optimization solution.

**Prompt template**: As a programming expert in the field of operations research and optimization, you offer programming examples and templates according to the background knowledge: $\{knowledge\}$. Now, according to problem description: $\{problem\}$. Could you please comprehend the extract code snippets in background knowledge and understand the their function, then give your code example to assist with addressing this problem. The comments given by your colleagues are as follows: $\{comments\}$, please refer to them carefully.

**Programming Expert:**

**Role description**: Skilled in programming and coding, capable of implementing the optimization solution in a programming language.

**Prompt template**: You are a Python programmer in the field of operations research and optimization. Your proficiency in utilizing third-party libraries such as Gurobi is essential. In addition to your expertise in Gurobi, it would be great if you could also provide some background in related libraries or tools, like NumPy, SciPy, or PuLP. You are given a specific problem and comments by other experts. You aim to develop an efficient Python program that addresses the given problem. Now the origin problem is as follow: $\{problem\}$ And the experts along with there comment are as follow: $\{comments\}$ Give your Python code directly.

**Code Reviewer:**

**Role description**: Conducts thorough reviews of the implemented code to identify any errors, inefficiencies, or areas for improvement.

**Prompt template**: As a Code Reviewer, your responsibility is to conduct thorough reviews of implemented code related to optimization problems. You will identify possible errors, inefficiencies, or areas for improvement in the code, ensuring that it adheres to best practices and delivers optimal results. Now, here is the problem: $\{problem\}$. You are supposed to refer to the comments given by your colleagues from other aspects: $\{comments\}$

### A.2.2 THE CONDUCTOR

The role of a conductor is highly specialized and significant, which necessitates a more intricate prompt design compared to other experts. The following is a prompt template for a conductor:

> You are a leader of an expert team in the field of operations research. Now, You need to coordinate all the experts you manage so that they can work together to solve a problem.
>
> Next, you will be given a specific OR problem, and your goal is to select the expert you think is the most suitable to ask for insights and suggestions.
>
> Generally speaking, the solution of a complex OR problem requires analysis, information extraction, modeling and programming to solve the problem. The description of problem is presented as follows: $\{problem\}$
>
> Remember, based on the capabilities of different experts and the current status of the problem-solving process, you need to decide which expert to consult next. The experts' capabilities are described as follows: $\{experts\_info\}$
>
> Experts that have already been commented include: $\{commented\_experts\}$
>
> REMEMBER, THE EXPERT MUST CHOOSE FROM THE EXISTING LIST ABOVE.
>
> Note that you need to complete the entire workflow within the remaining $\{remaining\_steps\}$ steps.
>
> Now, think carefully about your choice and give your reasons.

### A.2.3 THE REDUCER

The Reducer's role is to serve as a summarizer for all comments provided by the selected experts. They must meticulously analyze the comments and generate the final answer, which can take various forms, such as a modeling or a program. The Reducer's prompt template may vary based on the specific type of final answer required. Here's an example of the Reducer's prompt template when the goal is to obtain a program:

> You have been assigned the critical task of generating a program to solve the complex operations research problem presented. This program should incorporate the insights and suggestions provided by the selected experts. Your role is to synthesize the information effectively to create a functional program.
>
> The program is described as follows: $\{problem\}$
>
> The comments from other experts are as follows: $\{comments\}$
>
> Could you please write Python GUROBI code according to the comments.

## A.3 BASELINES' IMPLEMENTATION

To ensure a fair comparison, we have implemented the baseline algorithms following their original papers' guidelines.

The traditional model, **tag-BART**, typically requires a training process. If we were to directly use a pretrained tag-BART model from the LPWP dataset to test on the ComplexOR dataset, there would likely be a domain shift. To mitigate this issue, we can adopt a two-step approach. First, we pretrained the tag-BART model on the LPWP dataset. This initial pretraining enables the model to acquire basic NER abilities. Next, we fine-tune the pretrained model on an additional set of 30 problems that are similar to the ComplexOR problems. These problems have the same annotated format as the LPWP dataset. By fine-tuning the model on this specific set of problems, we aim to maximize the performance and adapt the model to the requirements of the ComplexOR domain.

For the **Standard** prompting technique, we leverage the in-context learning ability of the language model. Following the recommended approach outlined in the OpenAI official documentation, we design the following prompt template:

> You are a Python programmer in the field of operations research and optimization. Your proficiency in utilizing third-party libraries such as Gurobi is essential. In

addition to your expertise in Gurobi, it would be great if you could also provide some background in related libraries or tools, like NumPy, SciPy, or PuLP. You are given a specific problem. You aim to develop an efficient Python program that addresses the given problem. Now the origin problem is as follow:$\{problem\}$. Give your Python code directly.

**Chain-of-Thought** is a technique similar to standard prompting but with some additional steps. It begins with the sentence "Let's think step by step" to guide the model's thought process. After that, a further summarization step is added because the output generated by Chain-of-Thought can be lengthy and fragmented.

For **Tree-of-Thoughts** and **Graph-of-Thoughts**, we set the parameters based on the experiments conducted in the respective papers. The width of exploration is set to 3, and the maximum exploration step is set to 5. We adopt the prompt paradigm proposed in the work by Tree-of-Thoughts Prompting (Hulbert, 2023). The prompt is designed as follows, where $\{exploration\_step\_prompt\}$ represents the original prompt used in each exploration step:

Imagine three different experts in the field of operations research and optimization are modeling and writing programmer for a hard problem.

All experts will write down 1 step of their thinking, then share it with the group.

Then all experts will go on to the next step, etc.

If any expert realises they're wrong at any point then they leave.

The problem description is: $\{problem\}$

$\{exploration\_step\_prompt\}$

For **Progressive-Hint Prompting**, the original implementation may not be suitable for complex OR problems. In the original paper, the answer is an immediate number, which makes it easy to determine consistency between multiple interactions. However, in complex OR problems, the answer can be a model or a program, both of which are not directly comparable. To address this, we follow the underlying idea of Progressive-Hint Prompting and make some modifications. We generate an initial answer and then use an additional interaction with the language model to ask whether the current answer is the same as the previous one. In this way, the PHP algorithm is implemented in a more appropriate manner.

In the **ReAct** approach, there are two main steps: reasoning and acting. In the reasoning step, we use the same prompt as in CoT to guide the model's thought process. In the acting step, we limit the actions to retrieving knowledge from a knowledge base. This is because, in complex OR problems, accessing and utilizing external knowledge is crucial for making informed decisions. To ensure a fair comparison, we allow the ReAct agent to access all the knowledge bases mentioned in Chain-of-Experts.

The design of **Reflexion** aligns with the backward reflection process described in Chain-of-Experts. In Reflexion, feedback is obtained from the compilation and runtime of the modeling program, allowing for iterative refinement of the previous steps until the agent is satisfied with the answer. It's worth noting that in our experiment setting, we do not generate test units.

## A.4 MORE EXPERIMENTAL RESULTS

### A.4.1 DETAILED EXPERIMENT RESULT ON COMPLEXOR

Table 6 presents a detailed overview of the performance of baseline algorithms and the Chain-of-Experts approach applied to the ComplexOR dataset. In the interest of brevity, we employ abbreviations, referring to the main content of the traditional algorithm as "BART". Additionally, we use shorthand labels for various algorithms. The results highlight the challenges faced by both traditional algorithms, such as BART, and prompting techniques like CoT, as they were unable to successfully address all the problems in the dataset. Notably, GoT achieved success in solving the relatively straightforward "Blending" problem. Among the methods employing Large Language Models agents, which include ReAct and Reflexion, several less complex problems were solvable,

Table 6: Detailed experiment results for different methods on ComplexOR

| problem | BART | Standard | CoT | PHP | ToT | GoT | ReAct | Reflexion | SPP | CoE |
|---|---|---|---|---|---|---|---|---|---|---|
| Blending | ✗ | ✗ | ✗ | ✗ | ✗ | ✓ | ✓ | ✓ | ✗ | ✓ |
| Car Selection | ✗ | ✗ | ✗ | ✗ | ✗ | ✗ | ✗ | ✓ | ✗ | ✓ |
| Capacitated Warehouse Location | ✗ | ✗ | ✗ | ✗ | ✗ | ✗ | ✗ | ✗ | ✗ | ✗ |
| Employee Assignment | ✗ | ✗ | ✗ | ✗ | ✗ | ✗ | ✗ | ✗ | ✗ | ✗ |
| Aircraft Landing | ✗ | ✗ | ✗ | ✗ | ✗ | ✗ | ✗ | ✗ | ✗ | ✗ |
| VRPTW Routing | ✗ | ✗ | ✗ | ✗ | ✗ | ✗ | ✓ | ✗ | ✗ | ✓ |
| Flowshop | ✗ | ✗ | ✗ | ✗ | ✗ | ✗ | ✗ | ✗ | ✗ | ✗ |
| Distribution Center Allocation | ✗ | ✗ | ✗ | ✗ | ✗ | ✗ | ✗ | ✗ | ✗ | ✗ |
| Aircraft Assignment | ✗ | ✗ | ✗ | ✗ | ✗ | ✗ | ✗ | ✗ | ✗ | ✗ |
| Traffic Equilibrium | ✗ | ✗ | ✗ | ✗ | ✗ | ✗ | ✗ | ✗ | ✗ | ✓ |
| Robot Arm | ✗ | ✗ | ✗ | ✗ | ✗ | ✗ | ✗ | ✗ | ✗ | ✗ |
| Largest Small Polygon | ✗ | ✗ | ✗ | ✗ | ✗ | ✗ | ✓ | ✗ | ✗ | ✓ |
| CFLP | ✗ | ✗ | ✗ | ✗ | ✗ | ✗ | ✗ | ✗ | ✗ | ✗ |
| Cut Problem | ✗ | ✗ | ✗ | ✗ | ✗ | ✗ | ✗ | ✗ | ✗ | ✓ |
| Diet Problem | ✗ | ✗ | ✗ | ✗ | ✗ | ✗ | ✗ | ✗ | ✗ | ✗ |
| Dietu Problem | ✗ | ✗ | ✗ | ✗ | ✗ | ✗ | ✗ | ✗ | ✗ | ✗ |
| Knapsack | ✗ | ✗ | ✗ | ✗ | ✗ | ✗ | ✗ | ✗ | ✗ | ✗ |
| Multi-commodity Transportation | ✗ | ✗ | ✗ | ✗ | ✗ | ✗ | ✗ | ✗ | ✗ | ✗ |
| PROD | ✗ | ✗ | ✗ | ✗ | ✗ | ✗ | ✗ | ✗ | ✗ | ✗ |
| Single Level Big Bucket | ✗ | ✗ | ✗ | ✗ | ✗ | ✗ | ✗ | ✗ | ✗ | ✓ |
| Overall | 0/20 | 0/20 | 0/20 | 0/20 | 0/20 | 1/20 | 3/20 | 2/20 | 0/20 | 7/20 |

but overall performance remained suboptimal. The Chain-of-Experts (CoE) approach demonstrated the highest success rate, solving 7 out of 20 problems, making it the most effective algorithm.

We also found that certain algorithms, particularly PHP and SPP, struggled with token limitations when confronted with complex problems. This token limitation issue not only hindered their performance but also incurred increased computational costs and inefficiencies in their execution. In contrast, the Chain-of-Experts approach incorporates three key strategies to mitigate token limitation-related errors. Firstly, the utilization of expert summaries significantly reduces memory stress by nearly 50%. Secondly, the adoption of a Conductor, as opposed to a round-robin approach, further reduces the maximum context, effectively halving the token usage. Lastly, the incorporation of a visible map led to a substantial reduction of approximately 70% in token consumption.

### A.4.2 ABLATION EXPERIMENT ON INDIVIDUAL EXPERT

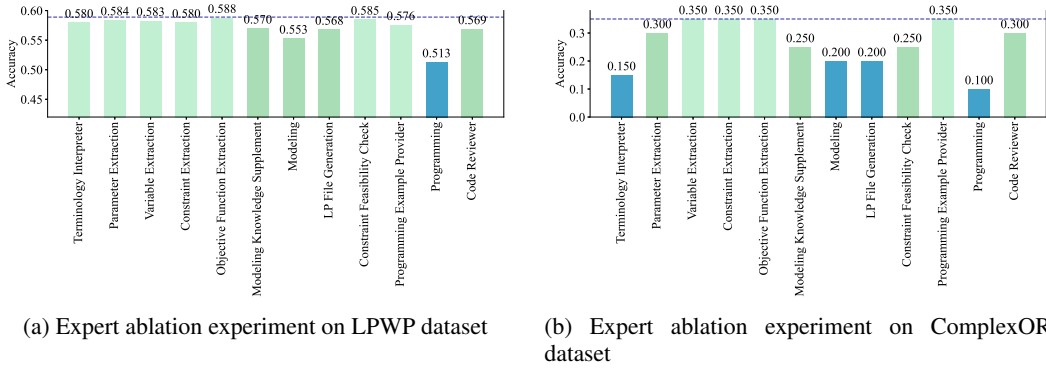

(a) Expert ablation experiment on LPWP dataset

(b) Expert ablation experiment on ComplexOR dataset

Figure 5: Impact on accuracy when removing individual experts from CoE

Figure 5 presents the results of the ablation experiment conducted on each expert in the Chain-of-Experts framework. The x-axis labels represent the removal of specific experts from CoE, while the y-axis represents the accuracy achieved after removing each expert. The blue horizontal line

represents the accuracy when all experts are integrated. Based on the results, we can observe the following insights regarding the importance of each expert for both datasets.

In the subfigure 5a, which corresponds to the LPWP dataset consisting of easy problems, removing a single expert does not lead to a significant performance drop. However, the most crucial expert is the Programming Expert. This finding aligns with the nature of the LPWP dataset, where the final evaluation is based on the correctness of the program. Therefore, having an expert who can provide insights into programming is essential. The second important expert is the Modeling Expert, as mathematical modeling plays a crucial role in problem-solving.

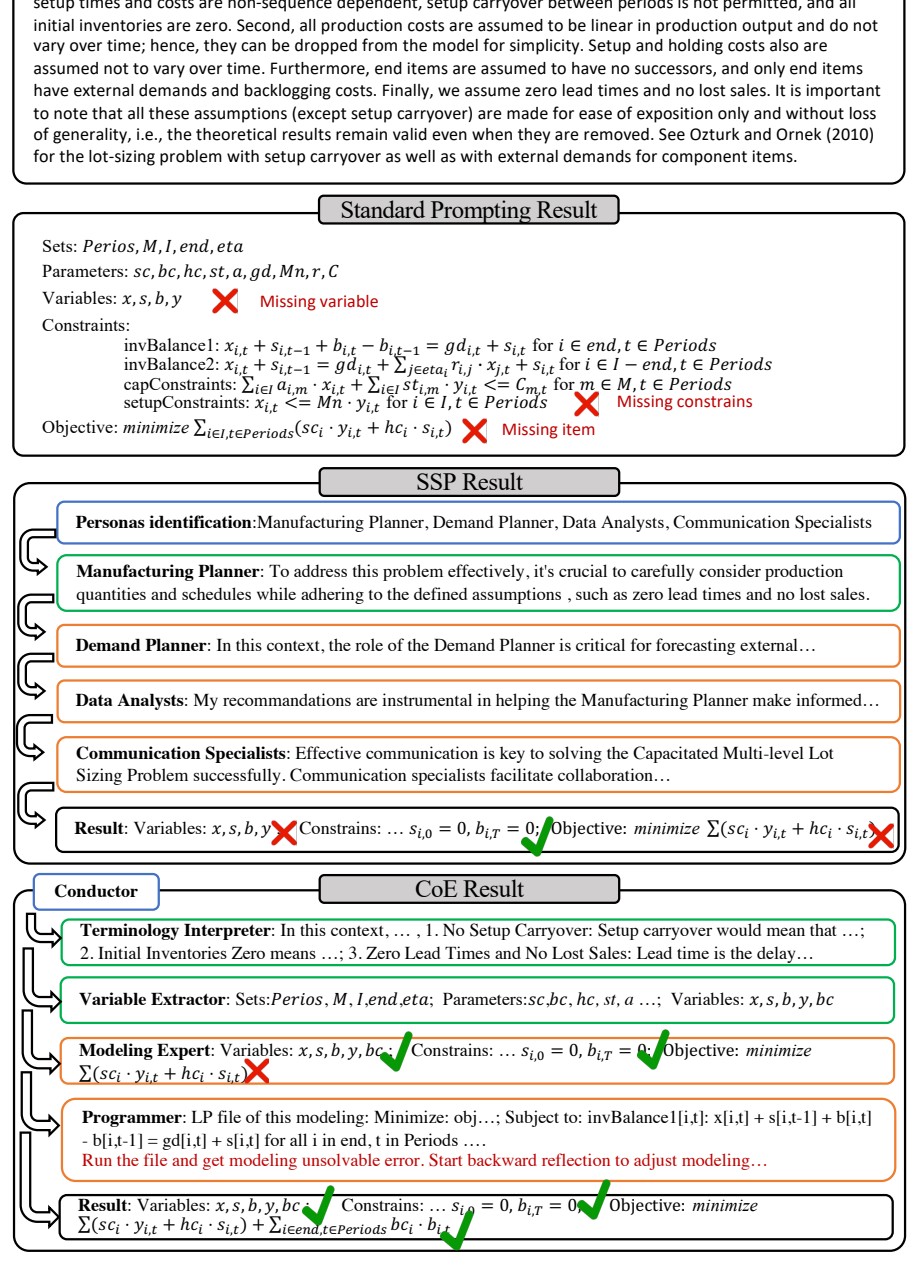

Figure 6: Case study

Subfigure 5b shows that individual experts have a much more significant impact on more challenging problems. Apart from the Programming Expert and Modeling Expert, the removal of the Terminology Interpreter leads to a significant drop of approximately $20\%$ in accuracy. This result highlights the knowledge-intensive nature of the ComplexOR dataset, which heavily relies on the supplementation of external knowledge. Interestingly, the LP File Generator Expert also proves to be important. This finding suggests that for harder problems, utilizing LP files as an efficient intermediate structural file for modeling is a good approach, as it yields better results compared to writing Python Gurobi program files.

### A.4.3 CASE STUDY

In this experiment, we conducted a detailed case study to gain insights into the effectiveness of our approach, as depicted in Figure 6. To reduce the uncertainty of sampling results, we run each methods five times and based our findings on the majority response. In this result, the standard prompting approach failed to correctly identify the variable $bc$ as backlogging cost, due to insufficient knowledge about the Multi-level Lot Sizing Problem. And there was a lack of constraints regarding initial and end statuses, which are essential in the context of real manufacturing process constraints. Moreover, the objective function was deemed unsolvable due to missing critical items. The SSP provided some basic background knowledge through a Manufacturing Planner created by leader persona. However, this method had limitations: three out of four personas offered negligible assistance in solving the problem, which is a critical issue in domain-specific problems like OR. In CoE, the Conductor effectively selected appropriate experts for different stages of the problem-solving process. Initially, a Terminology Interpreter was chosen to provide essential background knowledge. Although the Modeling Expert initially repeated the same mistake regarding the objective function, this error was rectified in the backward reflection process.

