# OpenReview forum: "Chain-of-Experts: When LLMs Meet Complex Operations Research Problems"
_ICLR.cc/2024/Conference — ICLR 2024 poster_

### Official Review · Reviewer_suKy · 2023-10-29

**Soundness:** 3 good
**Presentation:** 3 good
**Contribution:** 3 good
**Rating:** 6
**Confidence:** 3

**Summary:**

This work investigates solving complex operations research problems via the cooperation of multiple LLM-based agents.
The authors suggest Chain-of-Experts, which is a multi-agent framework comprised of 11 experts, for different aspects, and a conductor to coordinate these experts.
The experts are powered by common techniques such as In-context Learning and Reasoning based on LLMs.
The CoE framework sovle OR problems in an iterative way, where failed answers will get feedback via the reflection step.
This workflow will stop when the answer passes the evaluator or the iteration exceeds the given number.
A new benchmark, ComplexOR, is contributed to evaluate on 20 complex OR problems.
Experiments on LPWP and ComplexOR demonstrates that the proposed CoE outperforms previous LLM-agent methods.

**Strengths:**

1. the CoE framework.
2. A combination of existing techniques to solve OR problems.
3. A new small-scale real-world dataset

**Weaknesses:**

1. The results on ComplexOR seem not sense. A too small dataset.
2. The description of  CoE is not clear. It should be well-moviated and started with several backgrounds.

Through the response,  indeed the construction of ComplexOR is very difficult, and the authors acknowledge that the dataset will continue to be updated, which could be a potential contribution to the field and answer our questions.

**Questions:**

1. Is the CoE suitable for other reasoning tasks? What is the difference if applied to other tasks?
2. I suggest the paper give more attention to the CoE framework.

The answers have already addressed our questions.  Thanks.

---

> ### Author Response · Authors · 2023-11-21
> **Thanks for your comments!**
>
> We sincerely appreciate the constructive suggestions and comments on our work, which have definitely helped us to enhance the paper. The following are our detailed responses to each of the reviewer's comments.
>
> **Comment 1:**
>
> > The results on ComplexOR seem not sense. A too small dataset.
>
> We understand the reviewer's concern. ComplexOR can be viewed as the first dataset that has been well annotated for complex operations research problems and it has great potential to evaluate the reasoning capabilities of LLMs. The dataset is not large-scale because it is rather labour intensive for domain experts to help annotate the problems.  To construct an annotated problem instance, we need to
>
> 1. Find problems with suitable complexity from sources like academic papers, textbooks, or tutorials
> 2. Manually extract the textual description of the problem
> 3. Construct the mixed integer programming model and double check if it is correct
> 4. Programming for this model
> 5. Design several test cases with varied difficulty
>
> Since we are firmly aware of its value to the community, we continued to collect more samples after the paper submission. Till now, we have expanded the size of ComplexOR from 20 instances to 37 instances. The new experimental results on ComplexOR have been updated in the revision.
>
> **Comment 2:**
>
> > The description of CoE is not clear. It should be well-moviated and started with several backgrounds.
>
> In revision, we will follow the reviewer's advice to better motivate the description of CoE. Our thought started with Solo Performance Prompting, a multi-agent system. While SSP shows potential in solving complex tasks like logical grid puzzles, it falls short in more domain-specific taks such as the OR problem. We identified two primary reasons for this limitation. Firstly, in SSP, the agents are prompt templates initialized by leader personas, which restricts their ability to integrate extensive knowledge. To address this limitation, we introduce 'experts' to play specific roles in the problem-solving process. Secondly, we find that the collaborative approach in SSP is too straightforward to tackle challenging problems. It lacks the ability to selectively choose experts based on the specific problem and refine based on external feedback. To this end, we propose forward thought construction and backward reflection mechanisms inspired by real-world collaboration processes.
>
> **Comment 3:**
>
> > Is the CoE suitable for other reasoning tasks? What is the difference if applied to other tasks?
>
> This is a good point. In this paper, the agents/experts for collaborative reasoning are customized from the domain of operations research. As to agent design, the **terminology interpreter** who is responsible for supplying knowledge about specific terminology is integrated with a scenario modeling knowledge base. The **programming expert** is designed to generate modeling code, specifically in the LP file format.
>
> To extend CoE to support other reasoning tasks, the first step is to reconstruct proper agents for the target domain. Luckily, the collaborative reasoning framework among the agents with forward thought construction and backward reflection is re-usable.
>
> To evaluate the reusability of our framework, we conducted a preliminary experiment on Text2SQL task. Specifically, we selected 12 hard test instances related to a manufacturer scenario from the spider dataset[1]. In this experiment, we configured a CoE with two experts: a manufacturer terminology interpreter and a dataset grammar expert integrated with an SQLite grammar document[2]. We compared this setup with a standard prompting baseline. The results are as follows:
>
> |       | Execution Accuracy | Exact Matching Accuracy |
> | ----- | ------------------ | ----------------------- |
> | GPT-4 | 50%                | 37.5%                   |
> | CoE   | 75%                | 58.3%                   |
>
> We regret that time constraints prevented us from conducting more comprehensive experiments.   In our upcoming revisions, we are committed to adding a formal experiment to further validate our approach.
>
> **Comment 4:**
>
> > I suggest the paper give more attention to the CoE framework.
>
> In this work, we focus on automatically solving complex OR problems. It is motivated by the real demand of our industry partners. Such an automatic tool can greatly help them to reduce the overhead of modeling and programming for operations research problems from diversified industry sectors.
>
> We totally agree with your comment that it would generate more impact if we focus on CoE and apply it to more types of reasoning tasks.  In the previous comment, we have validated its potential to support complex Text2SQL task.  We believe this is an exciting avenue for future research to explore if we can further improve the generality of our reasoning framework among various reasoning tasks.
>
> Reference:
>
> [1] https://paperswithcode.com/sota/text-to-sql-on-spider
>
> [2] https://www.sqlite.org/lang.html

---

### Official Review · Reviewer_ocDx · 2023-11-01

**Soundness:** 3 good
**Presentation:** 3 good
**Contribution:** 3 good
**Rating:** 8
**Confidence:** 4

**Summary:**

This paper proposes chain of experts (CoE), a framework that uses multiple LLM agents to solve operations research (OR) problems. Most complex OR problem requires coordination among multiple experts, each solving a subproblem. In CoE, these experts are implemented using specialized LLMs augmented with, e.g., knowledge bases or reasoning skills targeting the subproblems they are designed to solve. A separate conductor model orchestrated this coordination process. This framework is further augmented by a backward reflection process, that, conditioning on the feedback provided by the program execution environment, recursively runs backward to identify potential errors in the chain. CoE does not require updating the parameters of the LLM agents, and thus is applicable to both proprietary and open-source models.

CoE is evaluated on LPWP (elementary linear programming problems), and complexOR (a newly created dataset by the paper, containing 20 expert-annodated OR problems). Experiments with GPT-3.5, GPT-4, and Claude-2 suggest that CoE outperforms baselines. An ablation analysis quantifies the contribution of each design choice in CoE.

**Strengths:**

- CoE is an interesting and novel framework for solving complex problems with multiagent collaboration.
- CoE’s design is grounded in real-world applications and proves effective.
- Requiring no training, CoE is applicable to both open-source and proprietary models.
- The presentation is reasonably clear.

**Weaknesses:**

- The paper would be more interesting to the LM community and have a larger impact if it could test out CoE on some of the well-established benchmarks
- ComplexOR is very small; I wonder how significant the results are
- The paper does not provide enough details on how the experts are specialized.

**Questions:**

- ComplexOR is very small. Can the authors provide more details on the consistency of the results across multiple runs?
- It would be interesting to compare to a baseline that applies CoE, and uses the same model to play all the different roles.
- Eq. 3 reads like applying the LLM to the prompt template outputs a new set of parameters, which does not align with what happens with prompting. At a higher level, do we really need the $\theta$ notations in Eqs. 2 and 3?

---

> ### Author Response · Authors · 2023-11-21
> **Thanks for your comments!**
>
> Thank you for your insightful comments. We have thoughtfully incorporated them into our revised paper. Below, we restate your comments and provide our point-by-point responses.
>
> **Comment 1: test out CoE on some of the well-established benchmarks**
>
> This is a good point. In this paper, the agents/experts for collaborative reasoning are customized from the domain of operations research. As to agent design, the **terminology interpreter** who is responsible for supplying knowledge about specific terminology is integrated with a scenario modeling knowledge base. The **programming expert** is designed to generate modeling code, specifically in the LP file format.
>
> To extend CoE to support other reasoning tasks, the first step is to reconstruct proper agents for the target domain. Luckily, the collaborative reasoning framework among the agents with forward thought construction and backward reflection is re-usable.
>
> To test the reusability of CoE, we conducted a preliminary experiment using the Text2SQL dataset. Specifically, we selected 12 hard test instances related to a manufacturer scenario from the spider dataset[1]. Here, we configured a CoE with two experts: a manufacturer terminology interpreter and a dataset grammar expert integrated with an SQLite grammar document[2]. We compared this setup with a standard baseline. The results are as follows:
>
> |       | Execution Accuracy | Exact Matching Accuracy |
> | ----- | ------------------ | ----------------------- |
> | GPT-4 | 50%                | 37.5%                   |
> | CoE   | 75%                | 58.3%                   |
>
> We regret that time constraints prevented us from conducting more comprehensive experiments.   In our upcoming revisions, we are committed to adding a formal experiment to further validation.
>
>
>
> **Comment 2: ComplexOR is too small**
>
> We understand the reviewer's concern. ComplexOR can be viewed as the first dataset that has been well annotated for complex operations research problems and it has great potential to evaluate the reasoning capabilities of LLMs. The dataset is not large-scale because it is rather labour intensive for domain experts to help annotate the problems.  To construct an annotated problem instance, we need to
>
> 1. Find problems with suitable complexity from sources like academic papers, textbooks, or tutorials.
> 2. Manually extract the textual description of the problem.
> 3. Construct the mixed integer programming model and double check if it is correct.
> 4. Programming for this model.
> 5. Design several test cases with varied difficulty.
>
> Since we are firmly aware of its value to the community, we continued to collect more samples after the paper submission. Till now, we have expanded the size of ComplexOR from 20 instances to 37 instances. The new experimental results on ComplexOR have been updated in the revision, as shown in Table 1 (overall performance), Table 2 (ablation study), and Table 3 (Robustness under different LLM).
>
>
>
> **Comment 3: a baseline that use the same model**
>
> This is an inspiring comment. We followed the reviewer's advice to implement a baseline that uses the same model. In this experiment, we use a uniform system prompt across all roles, without any additional knowledge bases. We also find that the removal of experts' features leads to a decrease in accuracy, which suggests that the CoE benefits from using specialized experts over a singular, generalized model. We have added this baseline in our revision.
>
> |                     |           | LPWP     |          |           | ComplexOR |          |
> | ------------------- | --------- | -------- | -------- | --------- | --------- | -------- |
> |                     | Accuracy  | CE rate  | RE rate  | Accuracy  | CE rate   | RE rate  |
> | CoE without Experts | 55.1%     | 4.0%     | 11.9%    | 18.8%     | 7.9%      | 15.0%    |
> | Chain-of-Experts    | **58.9%** | **3.8%** | **7.7%** | **25.9%** | **7.6%**  | **6.4%** |
>
>
>
> **Comment 4: role of theta in Eqs.2 and 3**
>
> We sincerely appreciate your comment and it actually pushes us to think about the roles of $\theta$ deeply. We agree that the use of $\theta$ in Equations 2 and 3 could be misleading as CoE don't actually generate new parameters through prompting. To clarify, we've revised our approach to align with the notation used in the paper Reflexion[3]. Here, we treat the Conductor as a policy, with its parameters indicated in the subscript. The revised equation is as follows:
>
> $$Conductor_{\mathcal{F}^{\theta'}(\mathbb{PT}_t)}(e | s) = P_r\{E_{\phi_t} = e | S_t = s\}$$
>
>
>
> Reference
>
> [1] https://paperswithcode.com/sota/text-to-sql-on-spider
>
> [2] https://www.sqlite.org/lang.html
>
> [3] Noah Shinn, Beck Labash, and Ashwin Gopinath. Reflexion: an autonomous agent with dynamic memory and self-reflection. CoRR, abs/2303.11366, 2023. doi: 10.48550/arXiv.2303.11366.

---

> > ### Comment · Reviewer_ocDx · 2023-11-22
> >
> > The authors' response has addressed most of my concerns. I have revised my review accordingly.

---

### Official Review · Reviewer_B8wx · 2023-11-01

**Soundness:** 3 good
**Presentation:** 3 good
**Contribution:** 3 good
**Rating:** 8
**Confidence:** 4

**Summary:**

This paper presents a multi-agent reasoning method for operations research problems solving. In particular, all the expert agents called in a sequence by another conductor agent, and all the agents are based on LLMs, acting different roles. The approach (named Chain-of-Experts) achieves better results compared with other SOTA models on the LPWP dataset and they also release a new dataset on complex OR problems.

**Strengths:**

* Propose a multi-agent method for OR problem solving with one conductor and multiple experts; and achieves better empirical results
* Release a dataset on complex OR for the community

**Weaknesses:**

* Lack of evaluation on individual expert agents, as well as the conductor
* The comparison with other models might not be fair, since they call the LLM differently. Maybe add some measurements of how different methods use the LLMs.

**Questions:**

* If we use other less competent LLMs, like smaller models or open sourced models, how much the performance will be affected?

---

> ### Author Response · Authors · 2023-11-21
> **Thanks for your comments!**
>
> We sincerely appreciate the constructive suggestions and comments on our work, which have definitely helped us to enhance the paper. The following are our detailed responses to each of the reviewer's comments.
>
> **Comment 1:**
> > Lack of evaluation on individual expert agents, as well as the conductor
>
> In our original submission, we have conducted an ablation study to evaluate individual expert agents. The reviewer can check Figure 5 in Appendix A.4.2. Due to space limit, we regret that it was not put in the main body of the paper.
>
>
> **Comment 2:**
> > The comparison with other models might not be fair, since they call the LLM differently. Maybe add some measurements of how different methods use the LLMs.
>
> In our experiments, we unify the way to use the LLMs. Each method within an experiment utilizes the same model API, facilitated by the 'chat complete' interface LangChain framework[1]. Please let us know if we did not interprete the reviewer's comment correctly.
>
>
> **Comment 3:**
> > If we use other less competent LLMs, like smaller models or open sourced models, how much the performance will be affected?
>
> This is a good angle. If we use smaller models such as Llama 2, the performance degrades dramatically. We have conducted an experiment in chapter 4.7 to show the sensitivity to different types of LLMs. For smaller models such as Llama 2, we also conducted additional experiments on dataset LPWP and presented the results in the following:
>
> |                  | Accuracy | CE rate | RE rate |
> | ---------------- | -------- | ------- | ------- |
> | Standard         | 0%       | 100%    | 0%      |
> | Chain-of-Thought | 0%       | 100%    | 0%      |
> | Reflexion        | 0%       | 100%    | 0%      |
> | ReAct            | 0.7%     | 99.3%   | 0%      |
> | Solo Performance | 0%       | 100%    | 0%      |
> | Chain-of-Experts | 0.7%     | 99.3%   | 0%      |
>
> In this experiment, we use the Llama2 13b model[2] as a foundational model, applying it across various baselines and the CoE on the LPWP dataset. Our findings revealed that the Llama2 model was largely ineffective in resolving the OR problems. Approaches like CoT and Reflexion showed no significant impact. Only 2 out of 289 instances were successfully resolved using ReAct and CoE, which incorporated external knowledge. These results suggest that Large Language Models must achieve a certain scale to effectively tackle complex problems.
>
>
>
> Reference
>
> [1] https://github.com/langchain-ai/langchain
>
> [2] Zhengliang Liu, Yiwei Li, Peng Shu, Aoxiao Zhong, Longtao Yang, Chao Ju, Zihao Wu, Chong Ma, Jie Luo, Cheng Chen, Sekeun Kim, Jiang Hu, Haixing Dai, Lin Zhao, Dajiang Zhu, Jun Liu, Wei Liu, Dinggang Shen, Tianming Liu, Quanzheng Li, Xiang Li: Radiology-Llama2: Best-in-Class Large Language Model for Radiology. CoRR abs/2309.06419 (2023)

---

### Official Review · Reviewer_CHCN · 2023-11-01

**Soundness:** 3 good
**Presentation:** 3 good
**Contribution:** 2 fair
**Rating:** 6
**Confidence:** 4

**Summary:**

This paper focuses on utilizing large language models (LLMs) to address operations research problems. It employs an approach where LLMs role-play as agents in the problem-solving pipeline, collaboratively breaking down and resolving problems. The paper also incorporates external feedback for the backpropagation reflections in the problem-solving pipeline, allowing the LLMs within the pipeline to self-improve. Moreover, the research introduces a new operations research dataset, which appears to be more intricate compared to existing ones. The proposed approach is tested on the newly-created dataset as well as another benchmark, and results indicate that it outperforms used baseline prompting methods.

**Strengths:**

1. Overall, the methodology presented in this paper is straightforward, easy to implement, and demonstrates strong empirical results across two benchmarks.
2. The paper offers a new operations research dataset that, based on experimental outcomes, is more challenging than existing ones.
3. I find the mechanism of propagating feedback from external sources to enhance the performance of language models both innovative and interesting. The results suggest that this mechanism also boosts model performance.

**Weaknesses:**

1. While the paper focuses on tackling complex operations research problems, it doesn't seem to introduce any techniques specifically tailored for operations research challenges.
2. I believe the novelty of this work is somewhat limited, as several studies have already explored the "planning with feedback" approach with LLMs. Please refer to "A Survey on Large Language Model based Autonomous Agents (https://arxiv.org/pdf/2308.11432.pdf)" for more details. I think the authors should offer a more in-depth comparison with these existing works. Moreover, though the methodology is described as a multi-expert framework, it essentially relies on deploying various prompts to the same LLM.

**Questions:**

Why can't the method proposed in this paper be represented through Solo Performance Prompting, and where exactly does it differ from Solo Performance Prompting? From the description, it seems that the approach is entirely representable under the Solo Performance Prompting framework.

---

> ### Author Response · Authors · 2023-11-21
> **Thanks for your comment!**
>
> We sincerely appreciate the constructive suggestions and comments on our work, which have definitely helped us to enhance the paper. The following are our detailed responses to each of the reviewer's comments.
>
> **Can CoE be Represented by the SSP Framework?**
>
> > Why can't the method proposed in this paper be represented through Solo Performance Prompting, and where exactly does it differ from Solo Performance Prompting?
>
> There are two key differences between Solo Performance Prompting and our CoE. First, in terms of agent generation, Solo Performance Prompting prompts a single LLM to identify and simulate multiple agents. In contrast, the agents/experts in CoE are tailored to collaboratively solve an OR problem. For example, the Modeling Knowledge Supplement Expert is integrated with the knowledge of games-cutting edge modeling to its knowledge base. Second, in terms of collaborative reasoning framework, the agents in Solo Performance Prompting cooperate in a round-robin manner. In contrast, our CoE works with forward thought construction and backward reflection.
>
> We also offer a case study in Appendix 4.3 to illustrate the differences of reasoning processes between Solo Performance Prompting  and our CoE.
>
>
>
> **Techniques Designed for OR Problems?**
>
> > While the paper focuses on tackling complex operations research problems, it doesn't seem to introduce any techniques specifically tailored for operations research challenges.
>
> This is a good point. In this paper, the agents/experts for collaborative reasoning are customized from the domain of operations research. As to agent design, the **terminology interpreter** who is responsible for supplying knowledge about specific terminology is integrated with a scenario modeling knowledge base. The **programming expert** is designed to generate modeling code, specifically in the LP file format.
>
> To extend CoE to support other reasoning tasks, the first step is to reconstruct proper agents for the target domain. Luckily, the collaborative reasoning framework among the agents with forward thought construction and backward reflection is re-usable.
>
> To test the reusability of CoE, we conducted a preliminary experiment using the Text2SQL dataset. Specifically, we selected 12 hard test instances related to a manufacturer scenario from the spider dataset[1]. Here, we configured a CoE with two experts: a manufacturer terminology interpreter and a dataset grammar expert integrated with an SQLite grammar document[2]. We compared this setup with a standard baseline. The results are as follows:
>
> |       | Execution Accuracy | Exact Matching Accuracy |
> | ----- | ------------------ | ----------------------- |
> | GPT-4 | 50%                | 37.5%                   |
> | CoE   | 75%                | 58.3%                   |
>
> We regret that time constraints prevented us from conducting more comprehensive experiments.  In our upcoming revisions, we are committed to adding a formal experiment to further validation.
>
>
>
> **Comparison with Other Planning with Feedback Methods**
>
> >  I think the authors should offer a more in-depth comparison with these existing works. Moreover, though the methodology is described as a multi-expert framework, it essentially relies on deploying various prompts to the same LLM.
>
> We thank the reviewer for providing the reference. The unique features of our CoE include forward agent selection and backward reflecion. We followed the reviewer's advice and provided an in-depth comparison with these existing works in the following table.
>
> |                  | Multi-agnets | Forward agent selection | External knowledge access | Refine by Feedback |
> | ---------------- | ------------ | ----------------------- | ------------------------- | ------------------ |
> | ReAct            | X            | X                       | ✓                         | X                  |
> | Voyager          | X            | X                       | ✓                         | ✓                  |
> | Ghost            | X            | X                       | ✓                         | ✓                  |
> | SayPlan          | X            | X                       | ✓                         | ✓                  |
> | MetaGPT          | ✓            | X                       | ✓                         | X                  |
> | NLSOM            | ✓            | X                       | ✓                         | X                  |
> | SSP              | ✓            | X                       | X                         | X                  |
> | ChatEval         | ✓            | X                       | X                         | X                  |
> | Chain-of-Experts | ✓            | ✓                       | ✓                         | ✓                  |
>
>
>
> Reference:
>
> [1] https://paperswithcode.com/sota/text-to-sql-on-spider
>
> [2] https://www.sqlite.org/lang.html

---

> > ### Comment · Reviewer_CHCN · 2023-11-23
> >
> > I have read the author's response, and some of my concerns have been alleviated. I've adjusted my scores accordingly.

---

### Author Response · Authors · 2023-11-21
**For all reviewers: further paper revision**

We will make the following revisions to the paper:

1. We further expand size of ComplexOR dataset from 20 to 37, and update the all experimental result related to ComplexOR. To improve coherence of result, we set the temperature of LLM to a value of $0.7$ and conduct five runs to average the metrics. The overall performance on ComplexOR is shown in following table and remaining results can be found in revision paper.

|                  | Accuracy  | CE rate  | RE rate  |
| ---------------- | --------- | -------- | -------- |
| tag-BART         | 0.0%      | -        | -        |
| Standard         | 0.5%      | 36.8%    | 8.6%     |
| Chain-of-Thought | 0.5%      | 35.3%    | 8.6%     |
| Progressive Hint | 2.2%      | 35.1%    | 13.5%    |
| Tree-of-Thought  | 4.9%      | 31.4%    | 7.6%     |
| Graph-of-Thought | 4.3%      | 32.4%    | 8.1%     |
| ReAct            | 14.6%     | 31.9%    | 10.8%    |
| Reflexion        | 13.5%     | 12.9%    | 10.1%    |
| Solo Performance | 7.0%      | 46.5%    | 13.5%    |
| Chain-of-Experts | **25.9%** | **7.6%** | **6.4%** |

2. To evaluate the effectiveness of introduction of expert, we wdd a baseline  that uses the same model, which uses a uniform system prompt across all roles without any additional knowledge bases.

|                     |           | LPWP     |          |           | ComplexOR |          |
| ------------------- | --------- | -------- | -------- | --------- | --------- | -------- |
|                     | Accuracy  | CE rate  | RE rate  | Accuracy  | CE rate   | RE rate  |
| Standard            | 42.4%     | 18.1%    | 13.2%    | 0.5%      | 36.8%     | 8.6%     |
| CoE without Experts | 55.1%     | 4.0%     | 11.9%    | 18.8%     | 7.9%      | 15.0%    |
| Chain-of-Experts    | **58.9%** | **3.8%** | **7.7%** | **25.9%** | **7.6%**  | **6.4%** |

3. Add an experiment that run methods in Llama2.

|                  | Accuracy | CE rate | RE rate |
| ---------------- | -------- | ------- | ------- |
| Standard         | 0%       | 100%    | 0%      |
| Chain-of-Thought | 0%       | 100%    | 0%      |
| Reflexion        | 0%       | 100%    | 0%      |
| ReAct            | 0.7%     | 99.3%   | 0%      |
| Solo Performance | 0%       | 100%    | 0%      |
| Chain-of-Experts | 0.7%     | 99.3%   | 0%      |

4. Modify the Equation 2 and 3, remove the notation of $\theta$ and the policy of Conductor is described as follows:
$$Conductor_{\mathcal{F}^{\theta'}(\mathbb{PT}_t)}(e | s) = P_r\{E_{\phi_t} = e | S_t = s\}$$

5. We provide a case study in Appendix 4.3 to illustrate insights into the effectiveness of our approach compared to standard prompting and Solo Performance Prompting.

---

### Meta-Review · Area_Chair_rwz5 · 2023-12-11

**Metareview:**

The papers study the use of LLMs for automatically approaching operations research problems. The authors propose chain-of-experts (CoE), in which a group of LLM agents cooperatively tackles the problem. In CoT, these agents can have different functionalities such as a conductor (which calls other agents, essentially a planner) and other agents with more specific roles. The evaluations in the paper are pretty thorough and the authors also put good effort during the discussion phase to address reviewer concerns. All reviewers and I agree that proposed framework is valuable and perform well compared to alternative approaches. While there are many prompting techniques, the proposed approach can be considered as reasonably novel and it represents a strong empirical/algorithmic contribution.

**Justification For Why Not Higher Score:**

Based on the average review score, this work can be bumped up to spotlight or even oral. I am currently recommending poster as I believe it is mostly based on intelligently prompting and coordinating LLMs and seems to lack a deeper sophistication.

**Justification For Why Not Lower Score:**

Reviewers unanimously agree on acceptance. The results are solid.

---

### Decision · Program_Chairs · 2024-01-16

Accept (poster)